# Compact Linear Flow Phantom Model for Retinal Blood-Flow Evaluation

**DOI:** 10.3390/diagnostics14151615

**Published:** 2024-07-26

**Authors:** Achyut J. Raghavendra, Abdelrahman M. Elhusseiny, Anant Agrawal, Zhuolin Liu, Daniel X. Hammer, Osamah J. Saeedi

**Affiliations:** 1Department of Ophthalmology and Visual Sciences, University of Maryland School of Medicine, Baltimore, MD 21201, USA; osaeedi@som.umaryland.edu; 2Center for Devices and Radiological Health (CDRH), U.S. Food and Drug Administration (FDA), Silver Spring, MD 20993, USA; anant.agrawal@fda.hhs.gov (A.A.);; 3Department of Ophthalmology, Harvey and Bernice Jones Eye Institute, University of Arkansas for Medical Sciences, Little Rock, AR 72205, USA; amelhusseiny@uams.edu

**Keywords:** retinal blood flow, adaptive optics, phantom, glaucoma, diabetic retinopathy, scanning laser ophthalmoscopy

## Abstract

Impaired retinal blood flow is associated with ocular diseases such as glaucoma, macular degeneration, and diabetic retinopathy. Among several ocular imaging techniques developed to measure retinal blood flow both invasively and non-invasively, adaptive optics (AO)-enabled scanning laser ophthalmoscopy (AO-SLO) resolves individual red blood cells and provides a high resolution with which to measure flow across retinal microvasculature. However, cross-validation of flow measures remains a challenge owing to instrument and patient-specific variability in each imaging technique. Hence, there is a critical need for a well-controlled clinical flow phantom for standardization and to establish blood-flow measures as clinical biomarkers for early diagnosis. Here, we present the design and validation of a simple, compact, portable, linear flow phantom based on a direct current motor and a conveyor-belt system that provides linear velocity tuning within the retinal microvasculature range (0.5–7 mm/s). The model was evaluated using a sensitive AO-SLO line-scan technique, which showed a <6% standard deviation from the true velocity. Further, a clinical SLO instrument showed a linear correlation with the phantom’s true velocity (*r*^2^ > 0.997). This model has great potential to calibrate, evaluate, and improve the accuracy of existing clinical imaging systems for retinal blood flow and aid in the diagnosis of ocular diseases with abnormal blood flow.

## 1. Introduction

Retinal blood flow is a crucial ocular biomarker for prevalent conditions that cause blindness worldwide. Dysregulation of retinal circulation has been linked to the pathogenesis of various ocular diseases, including age-related macular degeneration, glaucoma, and diabetic retinopathy [1,2,3,4]. Numerous studies have shown the association between retinal vascular alterations and systemic conditions such as hypertension, diabetes, stroke, and dementia [5,6]. Hanaguri and colleagues have demonstrated that retinal blood-flow regulation alterations may precede neural dysfunction in mice with type II diabetes mellitus [7].

Several ocular imaging techniques have been developed to measure retinal blood flow, such as optical coherence tomography angiography (OCTA), laser speckle imaging, laser Doppler velocimetry, laser Doppler flowmetry, and erythrocyte-mediated angiography (EMA) [8,9,10,11]. Among these, adaptive optics (AO)-enabled devices, which compensate for ocular aberrations and offer high lateral resolution, have enabled direct visualization of moving erythrocytes [12,13,14,15], the fine details of the vessel wall [4], and precise measurement of lumen diameter [16,17]. Burns et. al. first demonstrated an elegant method to capture red blood cell motion in space–time images using AO-SLO in line-scanning mode [18,19]. Bedggood and Metha described the use of an AO flood illumination system to assess erythrocyte velocity in retinal vasculature [13]. This approach enables the generation of high-contrast perfusion maps, offering detailed insights into the lumen of the retinal vascular network [20]. AO has also been combined with other imaging modalities, including OCT or EMA, to provide cellular-level resolution and additional blood-flow measurement capabilities [10,21]. Although more invasive, EMA enables the direct visualization of sparsely labeled indocyanine green (ICG)-loaded erythrocyte ghost cells using a clinical SLO instrument, providing precise quantification of ocular blood flow [22].

The quantification of blood-flow metrics from these imaging techniques often suffers from patient-specific variability and generalizability. Ocular imaging systems generally use the eye as an objective to visualize structures in the posterior segment. Consequently, flow rate measurements are influenced by normal variations in axial length, corneal curvature, and intraocular pressure, among other anatomical and physiological parameters [23,24,25]. Additionally, there is no universally accepted standard for acquiring absolute quantitative measures of blood flow, making it a challenge to compare results obtained across different ocular imaging devices.

Other major medical imaging modalities, such as X-ray computed tomography, magnetic resonance imaging, and ultrasound, use physical models known as “phantoms” to standardize system performance and optimize image quality. However, there is no accepted phantom for use in retinal vascular imaging. Several groups have developed retinal phantoms to assess structural imaging performance, mimicking the optical properties of the multi-layered retina [26,27,28]. For imaging flow, a phantom model has been created with microfluidic channels and an auxiliary pump in a circulation system to achieve various flow speeds [29]. However, their bulky setup, wet operation, and clogging issues in capillary-sized channels are potential disadvantages. A simple but reliable model is needed for the routine performance assessment of emerging clinical instruments that measure blood flow. Several groups have demonstrated that retinal blood-flow modelling demands multi-scale and multi-physics models to be physiologically realistic [30,31]. However, these theoretical predictions need validation with in vivo measurements, as demonstrated by Liu et al. [32]. Our work aims to aid in the functional assessment and calibration of in vivo imaging techniques.

In this study, we introduce a compact linear flow phantom based on a direct current (DC) motor and a conveyor-belt system. The phantom was validated on an AO-SLO device configured for line-scanning to measure retinal blood flow. Our flexible design allowed for the investigation of varying vessel angles with the AO-SLO line-scan technique. The phantom was also tested with a clinical SLO instrument configured for EMA to demonstrate its advantage in avoiding wet components and other complexities of alternate phantoms. 

## 2. Materials and Methods

### 2.1. Phantom Fabrication

A photograph and schematic of the assembled linear flow phantom are shown in Figure 1a,c. A high-torque 12V DC motor (Fuzhou Bringsmart Intelligent Tech. Co., Ltd., Fuzhou, China) with encoder and reduction gearbox (1:600) is coupled with a 6 mm wide timing belt via a GT2 synchronous timing pulley (12 mm diameter with 20 teeth). A 20 mm or 30 mm focal length achromat lens mounted in front of the belt serves as a focusing element, mimicking the human eye axial length range. The optical axis is orthogonal to a manual rotation mount (RSP1, Thorlabs), which can rotate the belt from 0 to 90°, allowing for operation over the full range of retinal vessel orientations. An idle pulley is centered with the optical axis and maintains constant distance between the belt and achromatic lens. A motor housing (Figure 1b) and supporting pieces were 3D-printed with Ultimaker 3 (Ultimaker BV, Utrecht, The Netherlands). A DC motor controller (1065B, Phidget Inc., Calgary, AB, Canada) and custom LabVIEW program (2018, National Instruments, Austin, TX, USA) were used to control the input voltage signal to the DC motor and monitor the rotational speed in real time at 10 Hz. The linear belt speed could be adjusted precisely within the range of physiological microvasculature flow velocities in the retina (0.5–7 mm/s) [33]. 

To mimic flowing red blood cells, polystyrene microspheres with a 10 μm nominal diameter (Cat# 24294-2, Polysciences, Warrington, PA, USA) were deposited on a flexible plastic slide to form a uniform monolayer using a drop-casting technique within a humid chamber, as shown in the schematic in Figure 2a. Near-infrared (IR) fluorescent beads (Degradex PLGA microspheres, CAT# LGFN10K, Sigma-Aldrich, St. Louis, MO, USA) were added to the polystyrene microsphere suspension in various number ratios motivated by the bead concentration ratios used in EMA imaging [34]. Bead density measurement was performed by Fiji (ImageJ version 2.3.0/1.54f) analysis of brightfield and fluorescence microscopy images taken with a 10× objective. Brightfield and fluorescence microscopy were both required to quantify the relative density ratios of non-fluorescent and fluorescent beads. Bead density assessment included at least ten different regions within each coated slide. After verification of the deposited beads on the slide under the micrscope, it was affixed to the belt surface, bead-side up, with double-sided tape. 

### 2.2. Phantom Calibration

The phantom was evaluated for bi-directional velocity linearity within the operational voltage (−12 V to +12 V) by observing the linear distance traveled by a marker on the belt with a PixelLink camera (PL-D7620MU-T) operating at 120 frames per second. Calibration curves were generated with the complete phantom assembly, which was used as the true linear velocity for all subsequent experiments. System repeatability was tested by measuring linear velocities for up to 5 s under continuous conveyor-belt operation. 

### 2.3. Phantom Evaluation

The Food and Drug Administration multimodal AO system is described in detail elsewhere [35]. Only the AO-SLO channel in line-scan mode was used to evaluate the phantom. The line-scan mode produces alternate raster and space–time images by intermittently holding the slow galvanometer scanner stationary [18]. The AO-SLO channel uses a superluminiscent diode (SLD, λ_c_ = 756 nm, ∆λ = 20 nm) for imaging. Another SLD (λ_c_ = 830 nm, ∆λ = 60 nm) was used as the Shack–Hartmann Wavefront Sensor (SHWS) beacon. A 30 μm diameter pinhole (1.5 Airy disc diameter) in the SLO detection path was used to resolve the beads via the confocal AO-SLO approach. The phantom was mounted on a three-axis translation stage, and optical alignment was performed with an adjustable iris for optimal positioning. The plastic slide with beads offered a diffuse reflection necessary to create a uniform array of bright spots on the SHWS, which was essential to achieve optimum wavefront correction. The specular reflection from the plastic microscope slide surface, which overwhelms the bead signal, was avoided by adding a slight tilt to the phantom optical alignment. Focus was set on the bead monolayer above the belt/slide surface. With a 1.5° × 1.5° FOV (raster mode), 100 interleaved frames were collected at 27 Hz with a resonant scanner frequency of 13.6 kHz in AO-SLO line-scan mode. During acquisition, the phantom belt was rotated by angle α relative to the fast-scan direction, which mimics vessel orientation in the retina and is required to create measurable streaks using the line-scan technique. Three videos were collected for each experimental setup to calculate measurement repeatability. A custom MATLAB (MathWorks, Natick, MA, USA) program was developed to derive the slope of the streaks (Δx/Δt) caused by individual beads on space–time images [36]. The velocity of individual beads was then computed as follows:v = (Δx/Δt) × sec(α),(1)

To investigate the effect of varying values of α on velocity calculation, videos were collected for each rotational stage position between 0 to 90° in steps of 10°, with all other parameters kept constant. After each modification, care was taken to check for focal plane and optimum wavefront correction to resolve the beads. The known focal length of the lens in air (*n* = 1) was used to convert the scan angle to the lateral distance at the focal plane using paraxial approximation.

### 2.4. Clinical SLO Imaging

The Heidelberg Spectralis HRA 2 (Heidelberg Engineering, Heidelberg, Germany), a clinical confocal SLO, was used in indocyanine green angiography (ICGA) mode with the excitation at 785 nm and the emission at 805–840 nm, values typically used in the EMA technique [34]. The phantom was mounted and optically aligned on the chin rest. A five-second video was collected with 15° FOV at 15.4 fps for each experimental setting. Collected video frames were further analyzed in Fiji (ImageJ version 2.3.0/1.54f) software. Instrument scaling output (µm/pixel) was used for image calibration. Automated bead tracking was performed for the velocity analysis using TrackMate, a Fiji tool developed by Tinevez and colleagues [37]. The algorithm detects beads in every frame using a Hessian feature detector. Frame-to-frame bead tracking was performed with a simple linear assignment problem (LAP)-tracker algorithm with optimized boundary conditions [38]. The mean velocity was then computed for each video by averaging at least twenty individual particle traces from the TrackMate results. All statistical analyses were performed in Excel (Microsof 365, Microsoft, Redmond, WA, USA) including mean, standard deviation (SD), Pearson correlation coefficient (*r*), and regression analysis. The *p*-value threshold was set to 0.05 for statistical significance.

## 3. Results

### 3.1. Calibration Curves

Figure 3a shows the linearity of the velocity measured with varying motor voltages from −12 to 12 V, where *r*^2^ > 0.999 with no load (*p* < 0.0001). After phantom assembly with the DC motor, the linearity was slightly lower, with *r*^2^ > 0.989 (*p* < 0.0001). Note that the DC motor required at least ±0.5 V to begin moving. The repeatability test showed SD < 4.8% (*n* = 3) at 6 V.

### 3.2. AO-SLO Imaging

For phantom evaluation, videos were collected from the AO-SLO channel in line-scan mode at different motor voltages within the linear range (single AO-SLO full frame shown in Figure 4a). In line-scan mode, space–time images contain streaks from the moving beads as shown in Figure 4b. A summary of mean velocities for various phantom velocity settings calculated from Equation (1) is shown in Figure 4c. Linear regression analysis showed a slope = 1.041 and *r^2^* value = 0.9983 (F_1,3_ = 1792; *p* < 0.05).

The phantom was further used to investigate the systematic effect of variations in vessel angles on AO-SLO line-scan accuracy. Using a constant belt velocity of 3.82 mm/s, we observed a non-linear increase in measured line-scan velocity above a 45° vessel angle, as shown in Figure 5a. Non-linear regression analysis with a cubic polynomial yielded the following best fit curve: y = 8.44 × 10^−5^x^3^ − 0.007x^2^ + 0.223x + 1.9 with *r*^2^ = 0.9934 (*p* < 0.05). Additionally, the observed streak length decreased with an increase in the vessel angle, as shown in Figure 5b.

### 3.3. Clinical SLO Imaging/EMA

To test our phantom with a clinical SLO instrument in ICGA mode, as used in EMA imaging, videos at 15.4 fps were collected while the phantom velocity was varied. Figure 6a shows a single SLO frame of the fluorescent beads with the phantom stationary. For videos collected at various phantom velocities, a maximum intensity projection of all video frames was taken to produce a single composite image, illustrating the bead motion in each video, as shown in Figure 6b. Phantom linear velocity thus creates clear trackable streaks from the fluorescent beads, where the distance between bead positions along the track can be used to compute the measured velocity. Results using TrackMate showed a linear relationship between the measured velocity and the true set phantom velocity with a slope = 0.9455 and *r*^2^ = 0.9975 (F_1,3_ = 1792; *p* < 0.05), as shown in Figure 6c. 

## 4. Discussion

In this study, we developed a simple and compact linear flow phantom for accurate retinal blood-flow velocity evaluation. The calibration curves showed excellent linearity after assembly, except for those close to the maximum voltage rating of the DC motor compared to no-load conditions. This difference is attributed to friction loss. Our phantom offered a linear velocity range between 0.5–7 mm/s, which corresponds to physiological flow velocities observed in the retinal microcirculation, and the modular design of our phantom allows for easy modifications for other ranges found in other non-biomedical applications. The bi-directional nature of the DC motor offered the ability to easily study different flow directions, as observed in arteries and veins in the retinal circulation. Our phantom, validated with AO-SLO and clinical SLO, achieves high accuracy with measured slopes close to unity. The low standard deviation of residuals for AO-SLO (0.1127 mm/s) and clinical SLO (0.1074 mm/s) demonstrates high precision.

For the AO-SLO line-scan approach, our phantom offers an optimal bead density for flow determination and was tested at various phantom velocities. We measured a linear correlation between phantom and AO-SLO derived mean velocities that was near unity. Previous reports showed that AO-SLO line-scan derived velocity measurement requires vessel selection such that the intersection angle between the vessel and scanner is always less than 45° for high accuracy [11,18]. Typically, a steeper streak and vessel angle results in reduced slope-determination sensitivity and shorter streaks on space–time images for a given flow velocity, respectively [39]. This happens because the time taken for erythrocytes to cross the scanning line is inversely proportional to the component of the erythrocyte velocity perpendicular to the fast scanning direction, as shown by Zhong et al. [18]. This limits the number of vessel segments where the flow can be quantified in any imaged region. With our phantom model, equipped with a flexible rotation stage, we systematically studied the effect of vessel angle on measured velocities, and showed a non-linear increase in error beyond 45°, up to ~400% error for angles near 90°. To correct this, a non-linear regression model fit to calibration data collected with the phantom could derive a correction factor for vessel angles between 40° to 90°. This could potentially extend the measurement of microvasculature flow rates to all vessels in an imaged region without any hardware modification. 

EMA offers direct quantitation of blood flow by tracing individual ICG-labelled red blood cells with a clinical SLO instrument. A recent study combined AO-SLO and EMA to quantify retinal blood-flow measurements in healthy human subjects [36]. The single-file flow observed with EMA was emulated with fluorescent beads, added at the ratio of 1:3 to the total beads, in our phantom and tested on a clinical SLO instrument. Results showed linear correlation between the phantom and clinical SLO-derived mean velocities, albeit with a slight underestimation. This deviation could be attributed to an error in the scaling calibration provided by the clinical SLO instrument, which assumes a human eye model compared to our phantom model in air. For accurate velocity measures in humans, it is imperative to know the imaged region’s true scaling (µm/pixel), which depends on the optics of the human eye. Many eye models have been proposed and used for the spatial correction of fundus images with varying degrees of complexity and parameterization [40], including axial length (AL) [41,42], lens thickness, refractive error [43], and anterior chamber depth [42,43]. Garraway-Heath et al. showed that accurate quantitation could be achieved using only AL [44]. Any deviation in AL from the true size could lead to an erroneous magnification factor and could impact critical clinical decisions in disease management [45,46,47]. 

Retinal phantoms play a crucial role in objective performance validation of imaging systems because subject variability is removed. Several groups have developed multilayer phantoms to emulate retinal structures [48,49,50]. However, most published phantoms are limited to structural assessment and do not mimic vascular blood flow. In another study, a biomimetic 3D-printed vascular phantom was developed via direct laser writing but lacked capillary single-file flow modeling [51]. Traditionally, circular polyethylene tubes or glass capillaries have been used in flow phantoms [52]. Although easy to fabricate, blood-vessel patterns and available lumen diameters are limited. Many microfluidic phantoms have been developed to accurately assess microvasculature and use intralipids to mimic flowing red blood cells with optimal concentrations. Materials like PDMS exhibit thermal shrinkage, which affects microfluidic channel lumen after curing and could lead to undesired flow patterns. In addition, bulky pump setups and underlying fluid dynamics add complexity to the phantom. Our phantom assembly uses off-the-shelf parts that are inexpensive, resulting in a total cost of less than $500, excluding the cost of the 3D printing and computer. While our phantom lacks anthropomorphic features, it avoids wet components and offers a simple, well-controlled model for accurate functional assessment. One of the limitations of our model is that the optical elements do not match the refractive index of the human eye. In addition, our model does not represent the vessel within retinal layers in depth. Furthermore, the current phantom only models relatively simple flow based upon the average blood velocity and does not consider many complex flow properties, hemodynamic effects, and physiological parameters that may impact actual retinal blood flow (red blood cell shape, Reynold’s number, blood as a multiphase non-Newtonian fluid, etc.). Some of these limitations could be addressed in future iterations of the phantom.

In the future, our phantom could be adapted for quantitative performance validation of both clinical and non-clinical imaging instruments. Additionally, the phantom is equipped to easily drive voltage signals in the software so that pulsatile blood flow could be mimicked to enable accurate calibration of dynamic parameters of retinal blood flow during the cardiac cycle. Our model can be used to calibrate, evaluate, and assess the velocity accuracy of current clinical imaging systems designed for retinal blood-flow measurement. Moreover, it can play a crucial role in assessing devices designed to characterize abnormal blood-flow patterns associated with many ocular diseases.

## 5. Conclusions

We presented a simple, compact, cost-effective, dry linear flow phantom for evaluating optical imaging devices that measure retinal blood velocity. The phantom model was optimized for bead density, including sparse fluorescent beads mimicking labeling in the EMA technique. The calibration showed a linearity in the 0.5–7 mm/s velocity range. Further, the model was calibrated and evaluated using a sensitive AO-SLO line-scan technique. The effects of varying vessel angles on AO-SLO results were systematically studied to improve the accuracy of retinal blood-flow quantitation. This model has great potential to calibrate and improve the accuracy of existing clinical imaging systems that measure retinal blood flow to aid in ocular disease diagnosis.

## Figures and Tables

**Figure 1 diagnostics-14-01615-f001:**
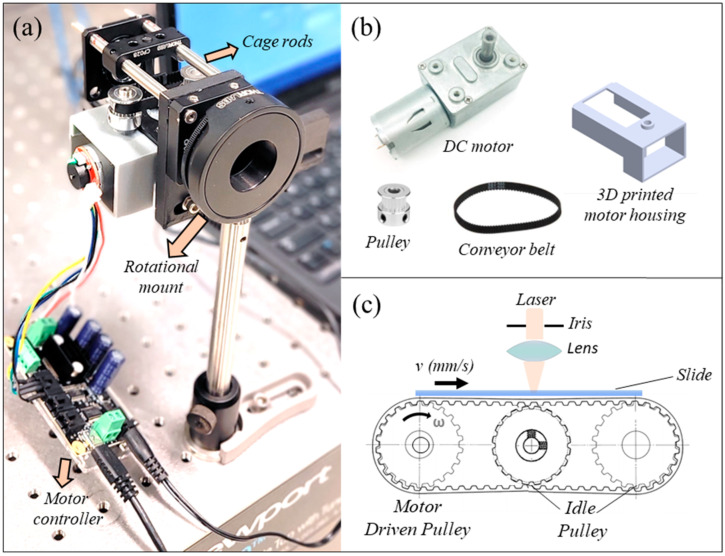
Linear flow phantom: (**a**) photograph of the phantom consisting of a motor and timing belt pulley system on a manual rotation mount attached to an optical post; (**b**) a high-torque DC motor with a synchronous timing belt, pulley, and 3D-printed parts used in the assembly; and (**c**) the schematic shows the optical system, including iris, lens, and timing belt on an idle pulley, such that the surface of the slide with beads is at the focal plane. The angular velocity of the pulley, ω °/s attached to the motor shaft, translates to linear velocity *v* mm/s.

**Figure 2 diagnostics-14-01615-f002:**
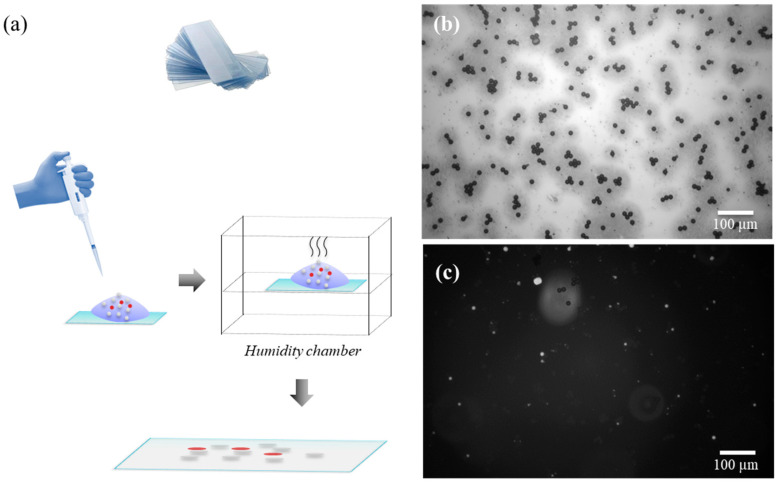
(**a**) Slide preparation for the phantom includes aqueous drop casting with an optimized concentration of polystyrene microspheres with fluorescent beads on a clean, flexible plastic slide. The preparation was dried overnight to achieve a uniform monolayer inside a humidity chamber. (**b**) White-light brightfield microscopy of the slide shows a monolayer mixture of fluorescent and non-fluorescent deposited beads with a density of ~580 particles/mm^2^. (**c**) The same region imaged with fluorescence shows the near-IR fluorescent beads only with a density of ~40 particles/mm^2^. Bead density measurement was performed using the ImageJ analyze particles tool at ten different regions on the slide.

**Figure 3 diagnostics-14-01615-f003:**
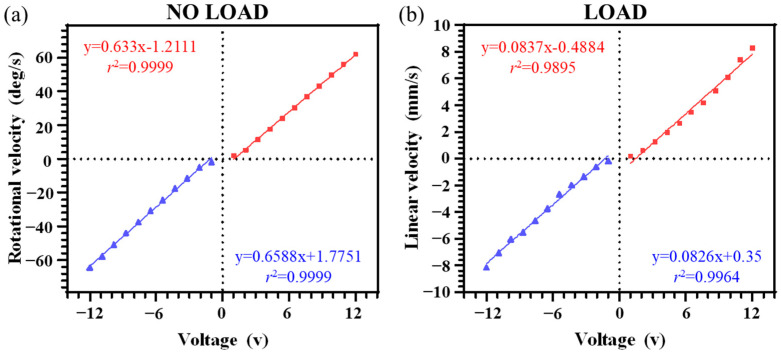
Performance of the phantom in the DC motor voltage range of −12 V to 12 V showed (**a**) a linear relation (*r*^2^ > 0.999) with rotational velocity (ω °/s) without load and (**b**) with complete phantom assembly showed a linear relation (*r*^2^ > 0.989) with belt linear velocity. Linearity for phantom use was measured between ±1.5 to 11 volts, corresponding to 0.5 to 7 mm/s belt linear velocity.

**Figure 4 diagnostics-14-01615-f004:**
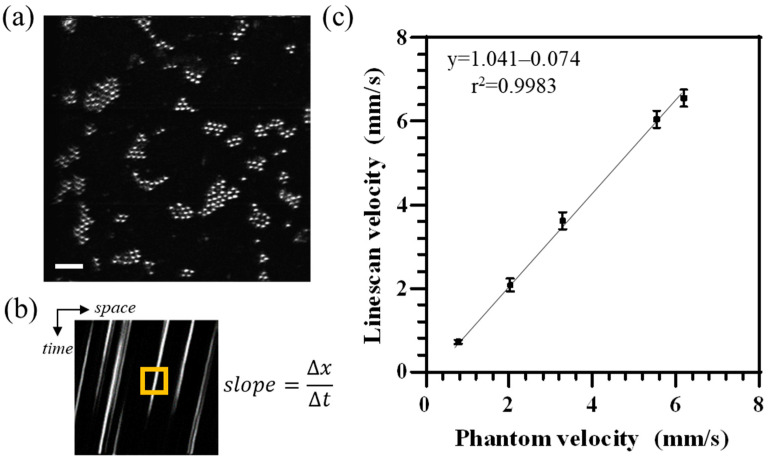
AO-SLO imaging of phantom with 20 mm focal length achromat. (**a**) A stationary AO-SLO raster image of beads. Scale bar = 50 μm. (**b**) A corresponding space–time image in AO-SLO line-scan mode shows streaks due to the motion of beads at a given phantom linear velocity. The orange box outline shows the sliding region of interest encompassing streaks, analyzed by the custom algorithm for slope calculation. (**c**) Mean velocity was computed for three independent video acquisitions for each input phantom velocity.

**Figure 5 diagnostics-14-01615-f005:**
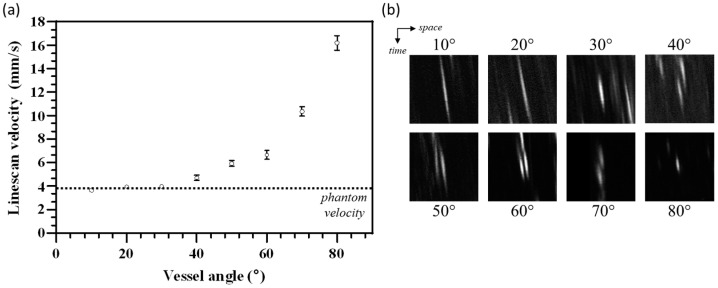
(**a**) The effect of varying vessel angle dependence in the line-scan approach is evident from the large error above 45°. The phantom used a 30 mm focal length achromat lens. Error bars indicated standard deviation over three independent measurements. (**b**) Streaks observed for each vessel angle setting at the same phantom velocity on space–time images show angle-dependent streak length.

**Figure 6 diagnostics-14-01615-f006:**
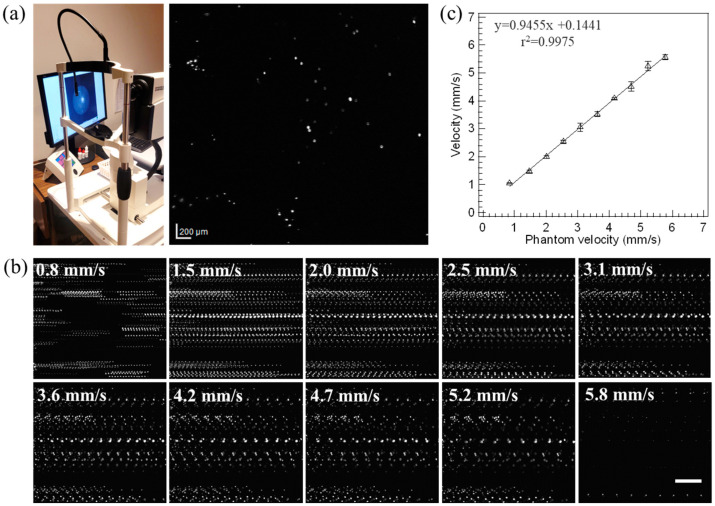
(**a**) Photograph of the clinical SLO used to test the phantom. A single frame from an ICGA video showed well-resolved fluorescent beads sampled at 15.4 fps; (**b**) a single composite image for each phantom velocity setting (0.8–5.8 mm/s) is produced by applying a maximum intensity projection across all frames in each video to visualize the motion traces of fluorescent beads. Scale bar is 200 µm; and (**c**) analysis of varying phantom velocity showed a linear trend (*r*^2^ > 0.997). Error bars indicate standard deviation over three independent measurements.

## Data Availability

The data presented in this study are available on request from the corresponding author.

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
