# Peer review of "Compact Linear Flow Phantom Model for Retinal Blood-Flow Evaluation"

_diagnostics, 2024, doi:10.3390/diagnostics14151615_

Round 1
Reviewer 1 Report
Comments and Suggestions for Authors
In this study, a linear flow phantom for evaluating retinal blood flow velocity was presented. The following corrections will enhance the MS:
As with other fields, computer-aided imaging studies have gained popularity in this area. For example, in the study with 10.1016/j.compag.2024.108963, identification and recognition were performed using retinal vessels, and in the study with 10.1016/j.pdpdt.2022.103157, disease diagnosis was conducted. Discuss in the discussion section how the proposed device will contribute to such studies.
Further interpretation and statistical analysis of the obtained results are required.
Please remove the blurriness from the text in Figures 3-5.
Reviewer 2 Report
Comments and Suggestions for Authors
In the last years the evaluation of current clinical imaging systems designed for retinal blood flow measurement is very important, especially in the investigation and treatment of different diseases and associated problems.
In this paper authors introduce new compact linear flow phantom based on a direct current motor and a conveyor belt system for the use in assessing devices designed to characterize abnormal blood flow patterns associated with ocular diseases. This device allowed investigation of varying vessel angles with the AO-SLO line-scan technique. In the manuscript it is demonstrated, that phantom is validated on an AO-SLO device configured for line-scanning to measure retinal blood flow and it is tested with a clinical SLO instrument configured for EMA, and the advantages in avoiding wet components are demonstrated. As it is mentioned by the authors, their device "has great potential to calibrate, evaluate, and improve the accuracy of existing clinical imaging systems for retinal blood flow and aid in diagnosis of ocular diseases with abnormal blood flow".
From the reviewer's point of view, this paper can be accepted after some minor revisions.
Comments:
1. In this field of research the crucial role is played by different mathematical and computer models. At first, why authors did not perform any theoretical investigations using mathematical models? At second, please mention some works on computational studies of such problems.
2. Please present the complete review of the previous papers in the Introduction with the detailed description of main results, presented in these works.
3. Please include some words on the properties of the considered blood flows: unsteady or quasi-steady, Reynolds number, the role of non-Newtonian effects, multiphase properties, average velocity, etc.
Reviewer 3 Report
Comments and Suggestions for Authors
The research work “Compact Linear Flow Phantom Model for Retinal Blood Flow Evaluation” explains the need for a well-controlled clinical flow phantom for standardization and to 21 establish blood flow measures as clinical biomarkers for early diagnosis and present a design and validation of a simple, compact, portable linear flow phantom based on a direct current motor and a conveyor belt system that provides linear velocity tuning within the retinal microvasculature range (0.5 - 7 mm/s). This evaluated using a sensitive AO-SLO line-scan technique, which showed <6% standard deviation from the true velocity. The work is well novel and well structured.
The comments are as follows:
1. Line 84: Our model can be used to calibrate, evaluate, and assess the velocity accuracy of 84 current clinical imaging systems designed for retinal blood flow measurement. Moreover, it can play a crucial role in assessing devices designed to characterize abnormal blood flow 86 patterns associated with many ocular diseases. Please avoiding claiming this in introduction section.
2. Line 127: (b) With white-light brightfield microscopy, the slide with deposited beads shows a monolayer (~580 particles/mm2), not able to understand what his image represent and explain.
3. Fig 5 (b) content need to explain in text.
4. Include a description of accuracy and precision of this model in brief.
5. Limitation of this model should be discuss/describe in detail.
6. In conclusion add the sentence about cost comparison.
